# 2-Deoxy-d-Glucose-Induced Metabolic Alteration in Human Oral Squamous SCC15 Cells: Involvement of N-Glycosylation of Axl and Met

**DOI:** 10.3390/metabo9090188

**Published:** 2019-09-17

**Authors:** Naeun Lee, Won-Jun Jang, Ji Hae Seo, Sooyeun Lee, Chul-Ho Jeong

**Affiliations:** 1College of Pharmacy, Keimyung University, Daegu 42601, Korea; rhee1218@naver.com (N.L.); mrdoin76@gmail.com (W.-J.J.); 2Department of Biochemistry, Keimyung University School of Medicine, Daegu 42601, Korea; seojh@kmu.ac.kr

**Keywords:** glycolysis, N-linked glycosylation, 2-deoxy-d-glucose, oral cancer, cancer metabolism

## Abstract

One of the most prominent hallmarks of cancer cells is their dependency on the glycolytic pathway for energy production. As a potent inhibitor of glycolysis, 2-deoxy-d-glucose (2DG) has been proposed for cancer treatment and extensively investigated in clinical studies. Moreover, 2DG has been reported to interfere with other biological processes including glycosylation. To further understand the overall effect of and metabolic alteration by 2DG, we performed biochemical and metabolomics analyses on oral squamous cell carcinoma cell lines. In this study, we found that 2DG more effectively reduced glucose consumption and lactate level in SCC15 cells than in SCC4 cells, which are less dependent on glycolysis. Coincidentally, 2DG impaired N-linked glycosylation of the key oncogenic receptors Axl and Met in SCC15 cells, thereby reducing the cell viability and colony formation ability. The impaired processes of glycolysis and N-linked glycosylation were restored by exogenous addition of pyruvate and mannose, respectively. Additionally, our targeted metabolomics analysis revealed significant alterations in the metabolites, including amino acids, biogenic amines, glycerophospholipids, and sphingolipids, caused by the impairment of glycolysis and N-linked glycosylation. These observations suggest that alterations of these metabolites may be responsible for the phenotypic and metabolic changes in SCC15 cells induced by 2DG. Moreover, our data suggest that N-linked glycosylation of Axl and Met may contribute to the maintenance of cancer properties in SCC15 cells. Further studies are needed to elucidate the roles of these altered metabolites to provide novel therapeutic targets for treating human oral cancer.

## 1. Introduction

Increased aerobic glycolysis and dependency on glycolysis for energy production are among the main metabolic hallmarks of cancer. Therapeutic strategies employing the metabolic differences between cancer and normal cells have been pursued as promising approaches for cancer treatment [1,2]. Unlike normal cells, cancer cells prefer aerobic glycolysis to oxidative phosphorylation, even in the presence of sufficient oxygen, to sustain their high proliferation rates [3]. This is known as the Warburg effect, and it contributes to cancer progression and emergence of drug resistance, thereby decreasing the efficiency of anticancer therapies [4]. Furthermore, enhanced glycolysis has been reported to promote angiogenesis and invasive tumor growth through acidification of the microenvironment [5,6]. Therefore, a better understanding of the mechanisms underlying the Warburg effect and its causes will lead to the development of more effective cancer treatments.

Various efforts have been made to develop anticancer treatments targeting tumor glycolysis [7]. Among them, 2-deoxy-d-glucose (2DG) has been clinically evaluated as a potent anticancer drug targeting glycolysis. 2DG is a structural analog of glucose whose hydroxyl group of the second carbon atom is substituted with hydrogen; thus, it cannot undergo glycolysis. Since 2DG interferes with the initial steps of the glucose metabolism, both glycolysis and oxidative phosphorylation are partially impaired [8]. This causes depletion of the cellular ATP level, blocked cell cycle, suppression of cell growth, and even cell death [9,10,11]. Combined administration of 2DG with other anticancer agents has been shown to be effective against xenografts of highly metastasized human cancers such as breast, osteosarcoma, and non-small cell lung cancer cells [12,13]. Although clinical trials have revealed limiting systemic toxicity of 2DG when used as a single agent, combined treatment with 2DG and other anticancer agents has been found safe and well-tolerated by patients in several phase I/II clinical trials [14,15,16,17,18].

Glycolysis blockage using 2DG has been shown to be effective in selectively killing tumor cells grown under hypoxia [19,20]. However, it has been argued that another mechanism instead of glycolysis blockage may be responsible for 2DG-induced cell death. In this context, it has been suggested that certain tumor cells grown under normoxia are sensitive to 2DG treatment due to the interference of 2DG with N-linked glycosylation rather than glycolysis [21]. Reports on the mechanism underlying the interference of 2DG with N-linked glycosylation are limited compared with those related to glycolysis. In particular, only limited studies have been conducted to elucidate the modulation of N-linked glycosylation of oncogenic receptors by 2DG [22,23]. Therefore, to achieve efficacious anticancer treatments, the action mechanisms of 2DG both in modulating glycolysis and N-linked glycosylation need to be elucidated.

Receptor tyrosine kinases (RTKs) constitute a large family of transmembrane glycoproteins that initiate downstream signaling cascades into the cell. Axl is a member of the Tyro2-Axl-Mer (TAM) family of RTKs whose activity is modulated by its ligand, growth arrest-specific gene 6 (Gas6) [24,25]. Axl regulates a diverse range of cellular responses including cell growth, survival, and metastasis, and its overexpression has been implicated in progression of several types of cancer including head and neck squamous cell carcinoma [26]. Met, another oncogenic RTK, is the only known high-affinity receptor for hepatocyte growth factor (HGF). Aberrant HGF/Met signaling has been shown to promote cancer cell invasion and metastasis in diverse cancer including oral squamous cell carcinoma (OSCC) [27]. Therefore, disrupting the function of the Axl and Met has been considered a therapeutic strategy for treating human cancer, but much is still unknown concerning their regulatory mechanisms and functions.

Here, to better understand the overall effects of 2DG on cell metabolism, we used mass spectrometry-based targeted metabolomics profiling in OSCC cell lines. OSCC is the most common neoplasm and a highly invasive malignant tumor characterized by a high rate of metastasis to the cervical lymph nodes [28]. In this study, we used SCC15 cells, which express highly activated oncogenic receptors, such as Met, and Axl as well as a key glycolytic regulator, HIF-1α, even under aerobic culture conditions. Accordingly, these cells are an appropriate cell line model to study the role of 2DG in cells that exhibit a glycolysis-dependent phenotype with highly activated oncogenic receptors. Thus, it is highly desirable to investigate the anticancer effect of 2DG and find a potential metabolic target for the development of anticancer therapeutics. Our results obtained here will help to elucidate the underlying action mechanism of 2DG and subsequent metabolic changes.

## 2. Results

### 2.1. DG-Induced Inhibition of Aerobic Glycolysis in Oral Cancer Cell Lines

To determine the metabolic differences in glycolysis, we examined the expression of glycolytic markers in two kinds of OSCCs. Notably, the mRNA levels of HIF-1α and LDHA in SCC15 cells was significantly higher than those in SCC4 cells, whereas the LDHB mRNA level was found to be lower in SCC15 cells (Figure 1A). Western blot analysis verified that the protein levels of these genes showed the same pattern (Figure 1B). To determine the effect of 2DG on aerobic glycolysis, we compared the levels of glucose consumption and lactate production in both cell lines. Our data revealed that SCC15 cells showed a much higher level of glucose consumption and lactate production than SCC4 cells (Figure 1C,D). Compared with the control groups, 2DG treatment decreased the lactate level and cell viability in both cell lines (Figure 1E,F). Moreover, colony formation assay showed decreased colony-forming ability of both cell lines upon 2DG treatment (Figure 1G). However, these effects of 2DG treatment were more pronounced on SCC15 cells, suggesting that SCC15 cells exhibit a much higher dependency on aerobic glycolysis and thus are more susceptible to 2DG treatment.

### 2.2. 2DG-Induced Interference of N-Linked Glycosylation of Axl and Met

Based on the notion that activation of RTKs results in enhanced glycolytic pathway in human cancers, we assessed whether the differential activities of the RTKs in these cell lines might cause the differential susceptibility to 2DG treatment. Toward this end, a human phospho-RTK array was performed. Interestingly, the phosphorylation levels of the RTKs, including Met, ErbB, and Axl in SCC15 cells were much higher than those in SCC4 cells (Figure 2A). Western blot analysis confirmed the higher phosphorylation levels of Met and Axl in SCC15 cells. (Figure 2B). Upon treatment with 2DG, these cells markedly reduced the expression of Met and Axl as well as HIF-1α, a master regulator of glycolysis (Figure 2C). Notably, the molecular sizes and expression levels of Met and Axl proteins decreased upon 2DG treatment in a dose-dependent manner (Figure 2C). The N-linked glycosylation inhibitor tunicamycin (0.01 μM) was used to confirm whether these changes in protein levels and sizes were due to the impairment of N-linked glycosylation. The results showed that these proteins were affected similarly as with the 2DG treatment, suggesting that N-linked glycosylation of these proteins was interfered by 2DG (Figure 2D). This result was further supported by the finding that the addition of mannose, but not pyruvate, reversed the 2DG-induced inhibition of N-linked glycosylation of Axl and Met (Figure 2E). However, treatment with pyruvate but not mannose reversed the 2DG-induced downregulation of HIF-1α, suggesting that HIF-1α expression might be regulated by glycolysis rather than N-linked glycosylation (Figure 2E). Next, to test whether the pharmacological inhibition of Axl and Met would be sufficient to inhibit cell viability, cell viability inhibition studies were performed with a selective Axl inhibitor, R428, and Met inhibitor, SU11274, respectively. Our data revealed that pharmacological inhibition of both Axl and Met reduced cell viability of SCC15 cells in a dose-dependent manner (Figure 2F). Also, RNA interference experiment showed that Axl and Met expression was reduced by treatment with siRNA against Axl and Met, respectively. As a result, there was corresponding decrease in cell viability (Figure 2G). Therefore, these data suggest that both Axl and Met play an important role in the survival of SCC15 cells, and these two receptors might contribute in part to the reduction of cell viability induced by 2DG.

### 2.3. Inhibition of N-Linked Glycosylation Contributes to the Anticancer Activity of 2DG in SCC15 Cells

To determine the relative contributions of N-linked glycosylation and glycolysis to the anticancer activity of 2DG, SCC15 cells were treated with 2DG in the presence or absence of mannose or pyruvate, and cell viability was assessed by the MTT assay. Our data revealed that the viability of SCC15 cells was reduced to 60% after treatment with 2DG alone, but recovered to approximately 80% when co-treated with 10-mM mannose (Figure 3A, right). However, the viability was only 68% with 10-mM pyruvate co-treatment (Figure 3A, left). Colony formation assay also showed that, unlike pyruvate, mannose significantly suppressed the anticancer effect of 2DG (Figure 3B). To determine how mannose suppressed the 2DG-induced reduction of cell viability, we examined the signaling molecules downstream of RTKs. Western blot analysis revealed that 2DG treatment decreased phosphorylation of AKT while increasing that of AMPK and this change was recovered by co-treatment with mannose rather than pyruvate (Figure 3C). Therefore, we concluded that interference with N-linked glycosylation rather than glycolysis might contribute to the anticancer effects of 2DG in SCC15 cells.

### 2.4. Metabolic Profiling by Targeted Metabolomics Analysis

To investigate the metabolic alterations induced by 2DG, (2DG + mannose), and (2DG + pyruvate), SCC15 cells were treated with 2DG in the presence or absence of mannose or pyruvate. Mass Spectrometry (MS)-based targeted metabolic analysis was performed using an AbsoluteIDQ p180 kit for the quantification of 187 metabolites, of which 103 metabolites were then selected by validation using the limit of detection (LOD) and limit of quantification (LOQ) cut-offs. To analyze the overall change in the composition of the metabolites in all the samples, we performed hierarchical clustering analysis. The samples were divided into four main groups and visualized using a dendrogram (Figure 4A). To obtain the key metabolites responsible for causing a significant difference among the groups, one-way ANOVA, partial least squares-discriminant analysis (PLS-DA), and significance analysis of microarrays (SAM) were used to examine 103 metabolites. Firstly, we conducted one-way ANOVA (FDR < 0.05) followed by a posthoc analysis using the Tukey HSD test for selection of the significant metabolites. Figure 4B shows these 66 significant metabolites marked with red circles. In the prediction model using PLS-DA, all the samples were clearly clustered into four groups (Figure 4C). In the PLS-DA model, the first principal component 1 (PC1), which contributed 42.2% to the separation of the four groups, was used to describe the difference among the groups. Generally, variable importance in projection (VIP) score (≥ 1) is used as a criterion for screening the highly contributing metabolites of each component in PLS-DA. Accordingly, 48 metabolites were selected; the top 25 metabolites are illustrated (Figure 4D). Finally, we used the SAM method to select the most discriminant and interesting metabolites. The delta value was set to 1.4 and FDR was 0.009, showing the good performance to select important features. As a result, a total of, 68 metabolites were identified (Figure 4E, green circles). Collectively, 66, 48, and 68 metabolites were selected by the independent three statistical methods and finally potential 44 metabolites were found common in all three statistical methods—PLS-DA, one-way ANOVA, and SAM (Table 1 and Figure 4F).

### 2.5. Important Altered Metabolites by 2DG

The results of the targeted metabolomics revealed that 44 potential metabolites consisted of 17 amino acids and biogenic amines, 22 glycerophospholipids, and 5 sphingolipids. Boxplots of the most significantly altered 17 amino acids and biogenic amines are presented in Figure 5. Interestingly, the levels of all the amino acids and biogenic amines were significantly decreased by 2DG treatment; however, this decrease was considerably recovered by co-treatment with pyruvate or mannose (Figure 5). In particular, mannose rather than pyruvate co-treatment recovered the amino acids, whose levels were comparable to those in the control (Figure 6A). Conversely, pyruvate was more effective than mannose in recovering the levels of the amines, such as AMDA and citrulline (Figure 5 and Figure 6B). Most sphingomyelins and phosphatidylcholines except for several phosphatidylcholines (aa C32:3, aa C34:3, aa C34:4, aa C36:6, and ae C40:1) were significantly upregulated upon 2DG treatment (Figure 7), and this effect was suppressed by pyruvate rather than mannose (Figure 6C,D).

## 3. Discussion

Although extensive studies on 2DG have focused on its function as a glycolytic inhibitor, it has been also reported that 2DG interferes with N-glycosylation of proteins. N-glycosylation is a co-translational modification, which is regulated by serial enzymatic reactions ending with the transfer of a core glycan from a lipid carrier to a protein substrate. Although the role of N-glycosylation in protein function has not yet been entirely elucidated, it is now being recognized as one of the most evident biological modification in tumorigenesis and metastasis [29,30]. In particular, N-linked glycosylation of RTKs has drawn attention because of their important role in tumorigenesis. Inhibition of N-linked glycosylation by tunicamycin has been shown to disrupt the RTK signaling in diverse tumor cells [31]. It has been reported that deglycosylation of Axl by tunicamycin inhibited metastasis of hepatocellular carcinoma cells to lymph nodes [32]. Additionally, molecular analysis of tunicamycin-treated tumors has revealed reduced levels of EGFR and Met in gliomas, suggesting that inhibition of N-linked glycosylation might be a novel therapeutic strategy against cancer [33]. In agreement with this, previous data suggest that glucose availability to the hexosamine pathway can regulate the surface expression of growth factor receptor in a manner dependent on N-linked glycosylation [34]. However, limited direct evidence on whether 2DG inhibits N-linked glycosylation of RTKs is available, necessitating further research [23]. Our data revealed that the two OSCC cell lines, SCC4 and SCC15, differed in sensitivity to 2DG. This observation could be explained by the following two reasons: First, SCC15 cells maintain a high level of HIF-1α expression even under normoxic culture conditions. Once activated, HIF-1α has been reported to reinforce the glycolytic phenotype by regulating the transcription of genes encoding most of the glycolytic enzymes [35]. Thus, SCC15 cells seem to produce energy by active aerobic glycolysis, as evidenced by high glucose uptake and lactate production. Second, SCC15 cells sustain their proliferation through the activation of oncogenic receptors. The phospho-RTK analysis revealed that RTKs, such as Axl and Met, were prominently activated in SCC15 cells, and the 2DG treatment induced deglycosylation of these proteins in a comparable effect to that of tunicamycin, which is a strong inhibitor of N-linked glycosylation. Indeed, 2DG inhibited N-linked glycosylation of these receptors, resulting in a decrease in cell viability and colony-forming ability. Therefore, aerobic glycolysis and N-linked glycosylation of Axl and Met in SCC15 cells are simultaneously inhibited by 2DG, and thus, the SCC15 cells are more sensitive to 2DG than SCC4 cells. These results suggest that 2DG may act as an effective anticancer drug to treat glycolysis-dependent cancers, which exhibit a higher activity of oncogenic receptors for their proliferation.

Metabolomics analyses of cancer cells or tissues quantitatively identify known or novel small molecular metabolites, and hence reflect the biological states of the cancer [36,37]. A comprehensive metabolomics analysis can detect even subtle changes in the concentrations of the metabolites. Such information can be used to identify targets for cancer diagnosis and treatment by integrating data analyses on metabolic changes with the existing transcriptome data. Additionally, recent advances in metabolomics have led to a better understanding of cancer metabolism, which has attracted much attention in cancer research but still remains relatively elusive. Therefore, several anti-cancer agents targeting metabolic processes in cancer have been developed and clinically investigated, including several glycolysis inhibitors, such as lonidamine, CPI-613, 2DG, and TLN-232. [38]. However, cancer-related metabolites altered by these drugs have not been probed by a metabolomics approach yet. Particularly, only few metabolomics approaches have been conducted to unravel the overall effect of 2DG metabolism in several human cancer cells [39,40,41].

Here, to assess the metabolic alterations triggered by 2DG in OSCCs, an MS-based targeted metabolomics approach was undertaken. We investigated how the metabolites in SCC15 cells were affected upon inhibition of glycolysis or N-linked glycosylation following 2DG treatment. To this end, pyruvate and mannose were used to suppress 2DG-induced inhibition of glycolysis and N-linked glycosylation, respectively. Our targeted metabolomics results showed that the levels of 44 metabolites were significantly changed by the 2DG alone and in combination with pyruvate or mannose. These 44 metabolites consisted of 17 amino acids and biogenic amines, 22 glycerophospholipids, and 5 sphingolipids. Among them, deregulation of the amino acids and biogenic amines was the most noticeable. Although the exact roles of amino acids in cancer metabolism are largely unexplored, numerous studies have emphasized the importance of specific amino acids, such as serine [42,43], glycine [44], alanine [45], proline [46], arginine [47], and asparagine [48,49]. Amino acids can act as important metabolic intermediates for cellular growth and maintenance, and in building blocks of proteins. Given that amino acids are consumed as essential nutrients in most cancer cells, it is expected that cancer cells have high concentrations of amino acids. Our results revealed that all the 17 amino acids and biogenic amines detected by targeted metabolomics existed at high concentrations in the control SCC15 cells and decreased upon 2DG treatment, which was counterbalanced by pyruvate or mannose treatment. In fact, a recent review has reported that amino acid levels are dysregulated in patients at the early stages of cancers, including lung, colon, and breast cancer [50]. Although there are some differences between different types of cancer, the concentrations of proline, histidine, phenylalanine, leucine, serine, glycine, proline, alanine, and threonine increase in both lung and breast cancer patients [51,52]. Consistent with this observation, our results also showed that these amino acids were downregulated in 2DG-treated SCC15 cells. Interestingly, the decrease in the levels of asparagine, serine, and threonine is notable since these amino acids are known as “consensus sequence of N-linked glycosylation” [53]. Moreover, our results confirmed a decreased putrescine level in 2DG-treated SCC15 cells, consistent with a previous study on human endometrial cancer cell lines [41]. Given that supplementation of the cell cultures with an amino acid mixture enhances glucose uptake and improves cell viability through glucose transporter type 4 [54], deregulation of amino acids might be closely related to the metabolic demand required for the proliferation of cancer cells.

However, it is not clear whether the metabolic alterations in most amino acids are directly related to the impairment of glycosylation of Axl and Met. Therefore, we cannot rule out the possibility that amino acid transporters other than RTKs could also be functionally altered by 2DG. Among these, a glutamine transporter ASCT2 (SLC1A5) can be a possible candidate as it was known to be responsible for intracellular transport of diverse amino acids such as glutamine, alanine, serine, asparagine, threonine, valine, and methionine [55]. Although there was no significant alteration in glutamine level among the 103 selected metabolites, recent reports suggest that ASCT2 may be a major cause in explaining the decrease in amino acid levels observed in this study. Indeed, ASCT2 has been shown to be glycosylated and its glycosylation might be critical for membrane localization and trafficking [56]. Moreover, other group also revealed the deglycosylation of ASCT2 upon glucose deprivation and 2DG treatment in leukemia cells [57]. Although it should be confirmed in future studies, it is thought that the deglycosylation of ASCT2 by 2DG and the alteration of metabolites are not irrelevant.

Our data also showed that the levels of most phosphatidylcholines and sphingomyelins were altered by 2DG in an opposite pattern to that of the amino acids. Although the biological relevance of this dysregulation is still unknown, it might be interpreted as a cytoprotective mechanism against 2DG treatment. Our results revealed that 2DG significantly upregulated several phosphatidylcholines, consistent with a former untargeted metabolic study that showed a global alteration in the lipid profile of human HaCaT keratinocytes upon 2DG treatment [39]. Several reports have indicated that sphingomyelin inhibits cancer progression [58,59]. In corroboration, the reduced cancer cell viability caused by 2DG treatment was accompanied by increased sphingomyelin levels, which may be an essential defense mechanism of dying cells against anticancer drugs. However, many of the mechanisms governing altered sphingolipid metabolism in cancer are yet to be fully understood.

In conclusion, our study suggests that inhibition of N-linked glycosylation of Axl and Met contributes to the anticancer effect of 2DG in SCC15 cells, and this modulation might be closely associated with the altered metabolite pattern identified by the targeted metabolomics. By combining metabolomics and biochemical analyses, our study lays the foundation for elucidating the anticancer effects of 2DG in detail, whereby exploitable targets for novel therapeutic strategies against human oral cancer can be identified.

## 4. Materials and Methods

### 4.1. Chemicals and Reagents

2DG, tunicamycin, SU11274, pyruvate, mannose, MTT, and dimethyl sulfoxide (DMSO) were obtained from Sigma-Aldrich (St. Louis, MO, USA). R428 was purchased from MedChemExpress (Monmouth Junction, NJ, USA). Antibodies against EGFR, p-Axl (Tyr702), Axl, p-Met (Tyr1234/1235), Met, p-Akt (Ser437), Akt, p-AMPK (Tyr172), and AMPK were purchased from Cell Signaling Technology (Beverly, MA, USA), and HIF-1α antibody was purchased from BD (BD Biosciences, Franklin Lakes, NJ, USA). Antibodies specific for GAPDH, LDHA, and LDHB, as well as the secondary antibodies, were purchased from Santa Cruz Biotechnology (Santa Cruz, CA, USA).

### 4.2. Real-Time PCR

Total RNA was isolated from SCC4 and SCC15 cells using TRIzol^TM^ reagent (Invitrogen, USA) according to the manufacturer’s instructions. Complementary DNA (cDNA) was synthesized from 2 μg total RNA using M-MLV reverse transcriptase (Promega, Madison, WI, USA). Real-time PCR amplification was performed with a LightCycler 480 (Roche) using double-stranded DNA-binding dye SYBR Green capillary mix AB gene system (Takara, Shiga, Japan). The comparative threshold (Ct) method was used to determine the target mRNA expression normalized with the Ct value of glyceraldehyde-3-phosphate dehydrogenase (GAPDH). The primer sequences used for target gene amplification are shown in the Table 2 below:

### 4.3. Phospho-RTK Array

Phospho-RTK array analysis was performed using a human phosphor-RTK array kit (R&D Systems, Minneapolis, MN, USA). Briefly, SCC4 and SCC15 cells were distributed at a density of 1 × 10^6^ cells/100 mm dish and incubated for 24 h. The cell lysates were prepared using lysis buffer containing the protease-phosphatase inhibitor cocktail. After the arrays were blocked with Array Buffer 1, they were incubated with 500-μg protein lysates 4 °C overnight. The arrays were then washed and incubated with an HRP-conjugated phosphor-tyrosine detection antibody for 2 h. The levels of the phosphorylated receptor tyrosine kinases were detected by chemiluminescence using LAS-3000 (Fuji, Japan). The intensities relative to the negative control were evaluated using Image J software.

### 4.4. Cell Lines and Cell Culture

Human oral squamous cell carcinoma SCC4 and SCC15 cell lines were obtained from American Type Culture Collection (ATCC, Manassas, VA, USA) and maintained in Dulbecco’s modified Eagle’s medium/F12 (DMEM/F12, 1:1) supplemented with 10% fetal bovine serum (FBS, HyClone Laboratories, Logan, UT, USA) and 1% penicillin/streptomycin in a 37 °C humidified incubator containing 5% CO_2_.

### 4.5. Cell Viability Assay 

Cell viability was assessed by the 3-(4,5-dimethylthiazol-2-yl)-2,5-diphenyltetrazolium bromide (MTT) assay. Briefly, at the end of the treatment, the culture medium was replaced with fresh DMEM/F12 medium supplemented with MTT (0.5 mg/mL MTT; 20 μL/well) and the cells were incubated at 37 °C for 4 h. After aspiration, 200 μL dimethyl sulfoxide was added, the plates were agitated, and absorbance was measured at 570 nm using FLUOstar Omega (BMG Labtech, Ortenberg, Germany).

### 4.6. Colony Formation Assay 

SCC4 and SCC15 cells were plated at a density of 800 cells/well in 6-well plates. After 2 weeks, cells were washed with Dulbecco’s phosphate-buffered, fixed with methanol for 2 min and stained with 0.1% crystal violet. Colony images were then captured, and the colony density was calculated using Image J software.

### 4.7. Western Blot Analysis 

SCC15 cells were seeded at a density of 1 × 10^6^ cells/100 mm dish. After 24 h of incubation, cells were treated with the indicated concentrations of 2DG in the presence or absence of chemicals such as tunicamycin, pyruvate, and mannose for 24 h. The cells were then lysed with cold whole-cell lysis buffer supplemented with Halt™ Protease and phosphatase inhibitor cocktail, PMSF, and EDTA, followed by sonication and centrifuged at 16,000× *g* for 20 min at 4 °C. Total protein levels were quantified using the BCA protein assay kit (Thermo Scientific, Rockford, IL, USA). Total protein samples (30 µg) were resolved using SDS-PAGE and transferred onto PVDF membranes (Millipore Corp., Bedford, MA, USA). The membranes were blocked with 5% BSA in TBS-T at room temperature (RT) for 1 h and then incubated with the specific primary antibodies at 4 °C overnight. The membranes were then washed with TBS-T for 3 times and incubated with the HRP-conjugated secondary antibody (1:10,000) at RT for 1h. The protein bands were visualized using SuperSignal^®^ West Dura Extended Duration Substrate (Thermo Scientific, Waltham, MA, USA) and developed with LAS-3000 (Fuji, Japan). 

### 4.8. RNA Interference 

Double-stranded siRNAs against con (sc-37007), Axl (sc-29769) were synthesized by Santa Cruz Biotechnology. Met (Cat. No.: 4233-1) siRNA was obtained from Bioneer (Daejeon, Republic of Korea). SCC-15 cells were seeded at 4 × 10^5^ cells/well in 60 mm dish. After overnight incubation, cells were transfected with siRNA of target genes or with control siRNA using Oligofectamine transfection reagent (Invitrogen, USA) according to the supplier’s protocol, followed by drug treatment. Gene silencing efficacy of siRNA was measured by Western blotting.

### 4.9. Measurement of Lactate Production and Glucose Consumption

SCC4 and SCC15 cells were seeded at a density of 1 × 10^5^ cells/well on 6-well dishes and incubated for 48 h in the presence or absence of 2DG. The media were collected and assessed for their glucose and lactate contents using the glucose and lactate assay kit (BioVision, Inc., Milpitas, CA, USA) according to the manufacturer’s instructions. The optical density (OD) values at different time periods were measured. Glucose consumption and lactate production were calculated based on the standard curve drawn from the OD values and normalized by total protein (µg) measured using the BCA protein assay kit.

### 4.10. Targeted Metabolomics Analysis

For targeted metabolomics analysis, SCC15 cell lysates were prepared as follows: Cells were washed with ice-cold PBS and lyzed in 10-mM phosphate buffer. Then, three cycles of sonication (40 kHz; 25 °C; 15 s) followed by a freeze-thaw cycle (liquid nitrogen for 30 s followed by instant thawing on a 98 °C heat block) were performed, and the samples were centrifuged to collect the cell lysates (20,000× *g*, 10 min, 4 °C). 

As we previously described [60], metabolic alterations were evaluated using an AbsoluteIDQ p180 kit (Biocrates Life Sciences AG. Innsbruck, Austria) which allowed the simultaneous quantification of 187 metabolites (40 acylcarnitines, 41 amino acids and biogenic amines, 90 glycerophospholipids, 15 sphingolipids, and 1 monosaccharide). An AB Sciex 4000 QTRAP^®^ Mass Spectrometer (Sciex, Framingham, MA, USA) was used for analysis in the multiple-reaction monitoring detection mode with electrospray ionization (ESI) at Inha University Hospital Clinical Trial Center (Incheon, Korea). Amino acids and biogenic amines were injected into the mass spectrometer using FIA, and the other groups of metabolites were injected via LC. The kit was validated using MetVal™ software (Biocrates Life Sciences AG), and the analytical results were processed using Analyst™ (version 1.6.2; AB Sciex) and MetVal™ software (Biocrates Life Sciences AG). The quantitative metabolite results were normalized to the total protein concentrations of the cell lysates. 

### 4.11. Data Processing and Statistical Analyses

A multivariate statistical analysis was performed using MetaboAnalyst (version 4.0) software, a web-based platform for data processing and interpretation (www.metaboanalyst.ca). The raw targeted metabolomics data were normalized with the median value and auto-scaled before statistical analysis. Unsupervised hierarchical clustering was performed for the natural extraction of similar groups among the samples. For clustering, the Euclidean distances were used as the similarity measure, and Ward’s linkage was used as the clustering algorithm. The result of clustering was visualized using a dendrogram. To assess the significance of class discrimination among the groups and selected important metabolites, partial least square discriminant analysis (PLS-DA) was performed among the four groups. PLS-DA model was validated using R2 and Q2 based on leave one out cross-validation (LOOCV). Variable importance in projection (VIP) score was used to determine potential metabolites that contributed to the PLS-DA separation (VIP score ≥ 1). One-way ANOVA and Tukey’s HSD test were used to identify the metabolites that significantly differed among all the groups using a false discovery rate (FDR) of q < 0.05. Significance analysis of microarray (SAM) was used to classify and select the key metabolites. The box plots of 44-significant metabolites from PLS-DA, ANOVA and SAM methods were displayed using Graphpad Prism.

## Figures and Tables

**Figure 1 metabolites-09-00188-f001:**
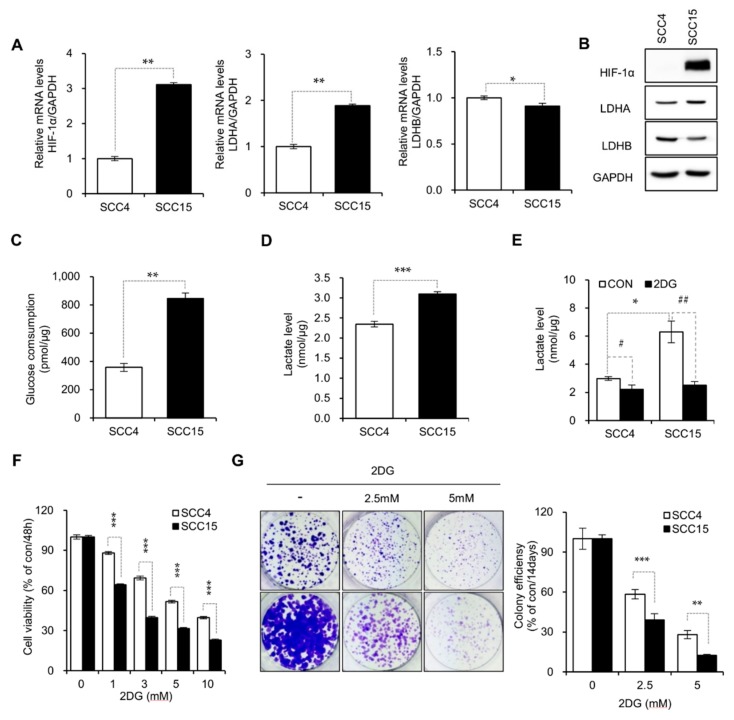
SCC15 cells are susceptible to 2DG treatment. (**A**) Quantitative real-time PCR measuring the levels of the indicated genes. GAPDH was used for Ct value normalization. Statistical significance was determined using the Student’s *t*-test. * *p* < 0.05; ** *p* < 0.01. Error bars show mean ± SD (*n* = 3). (**B**) Western blot analysis for the expression of LDHA, LDHB and HIF-1α proteins in SCC4 and SCC15 cells. GAPDH was used as a loading control. (**C**) Glucose consumption levels in SCC4 and SCC15 cells were measured 48 h after seeding. Error bars represent mean ± SD (*n* = 3). Statistical significance was determined using the Student’s *t*-test. ** *p* < 0.01. (**D**,**E**) Lactate levels in SCC4 and SCC15 cells were measured 48 h after seeding followed by treatment without (**D**) or with (**E**) 2DG. Error bars represent mean ± SD (*n* = 3). Statistical significance was determined using the Student’s t-test. * *p* < 0.05; *** *p* < 0.001 versus SCC4. # *p* < 0.05; ## *p* < 0.01 versus 2DG. (**F**) SCC4 and SCC15 cells were seeded at the density of 1.5 × 10^3^ cells/well in 96-well plates and treated with the indicated concentrations of 2DG for 48 h. Cell viability was determined using the MTT assay. Error bars show mean ± SD (*n* = 4). Statistical analysis was conducted using two-way ANOVA. *** *p* < 0.001, compared with SCC4 cells. (**G**) Colony formation assay. Cells were treated with 2.5-5 mM 2DG as indicated and stained with crystal violet after 14 d of incubation. The representative images from three independent sets of experiments (left) and the quantification graph of colony formation (right) are shown. Error bars represent mean ± SD (*n* = 3). Statistical analysis was conducted using two-way ANOVA. ** *p* < 0.01; *** *p* < 0.001, compared with SCC4 cells.

**Figure 2 metabolites-09-00188-f002:**
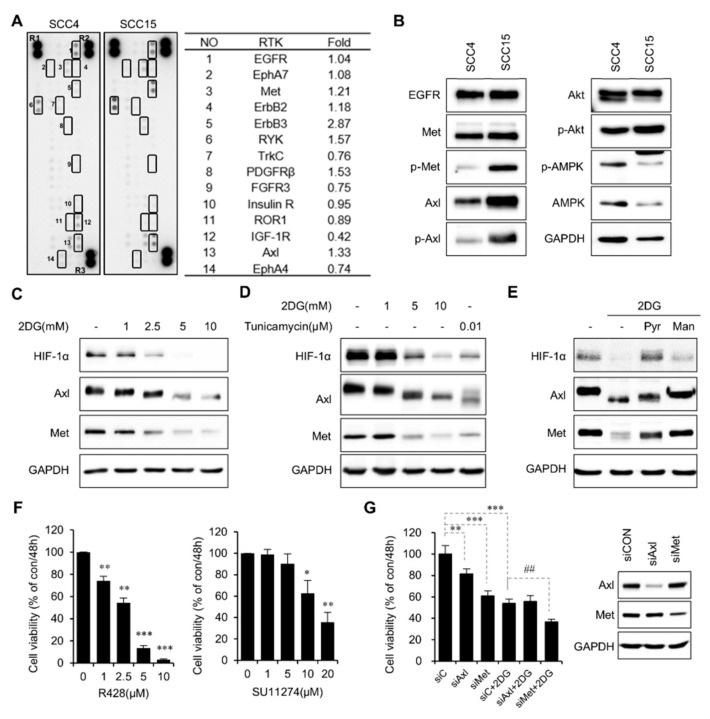
2DG interferes with N-linked glycosylation of Axl and Met in SCC15 cells. (**A**) Phospho-RTK analyses of SCC4 and SCC15 cells. Phosphorylation levels were quantified using ImageJ software and normalized to the reference spots (R1, R2, and R3). (**B**) Protein levels of several RTKs and downstream signaling proteins. Cell lysates were assayed using western blot with antibodies against RTKs and downstream proteins. GAPDH was used as a loading control. (**C**) Western blot analysis of target proteins after treatment with 2DG for 24 h. Axl, Met, and HIF-1α protein levels were measured; GAPDH was used as a loading control. (**D**) Western blot analysis of target proteins after treatment with 2DG or tunicamycin for 24 h. Axl, Met, and HIF-1α protein levels were measured; GAPDH was used as a loading control. (**E**) Western blot analysis of target proteins after treatment with 2DG in the presence or absence of pyruvate or mannose for 24 h. Axl, Met, and HIF-1α protein levels were measured; GAPDH was used as a loading control. (**F**) SCC15 cells were treated with R428 (left panel) or SU11274 (right panel) for 48 h, and cell viability was measured by the MTT assay. The values are expressed as a percentage of the control. Error bars represent mean ± SD (*n* = 3). Statistical significance was determined using the Student’s *t*-test. * *p* < 0.01, ** *p* < 0.01, *** *p* < 0.001 versus control. (**G**) SCC15 cells were treated with 5 mM 2DG after knockdown of Axl or Met with each siRNAs for 24 h, and cell viability was measured by the MTT assay. The values are expressed as a percentage of the control. Error bars represent mean ± SD (*n* = 3). Statistical significance was determined using the Student’s *t*-test. ** *p* < 0.01, *** *p* < 0.001 versus siC. ## *p* < 0.01 versus siC + 2DG.

**Figure 3 metabolites-09-00188-f003:**
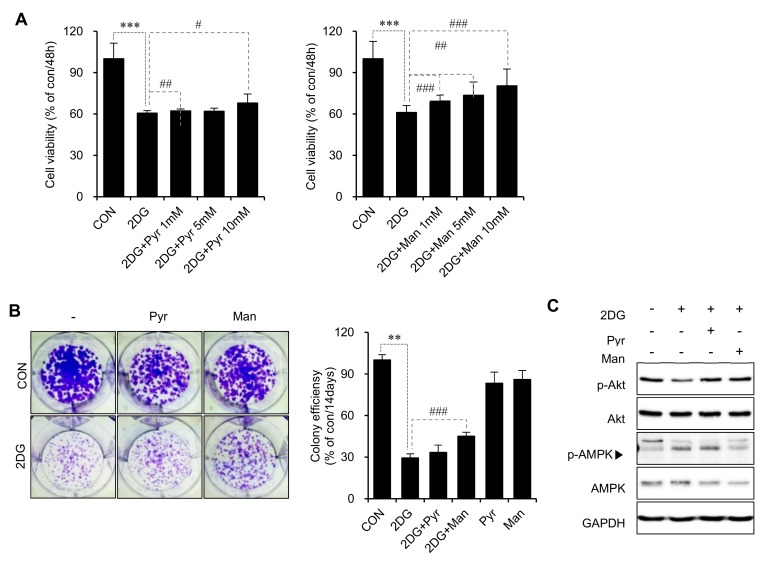
Inhibition of N-linked glycosylation contributes to the anticancer effect of 2DG in SCC15 cells. (**A**) SCC15 cells were treated with 2.5 mM 2DG in the presence or absence of pyruvate (left panel) or mannose (right panel) for 48 h, and cell viability was measured by the MTT assay. The values are expressed as a percentage of the control. Error bars represent mean ± SD (*n* = 4). *** *p* < 0.001 versus CON. # *p* < 0.05; ## *p* < 0.01; ### *p* < 0.001 versus 2DG. (**B**) Colony formation assay. Cells were treated with 2DG in the presence or absence of pyruvate or mannose as indicated and stained with crystal violet after 14 d of incubation. Error bars represent mean ± SD (*n* = 3). Statistical significance was determined using the Student’s t-test. ** *p* < 0.01 versus CON. ### *p* < 0.001 versus 2DG. (**C**) SCC15 cells were treated with 2DG in the presence or absence of pyruvate or mannose for 24 h, and protein expression was measured by western blot analysis. The phosphorylated and total levels of AKT and AMPK were measured; GAPDH was used as the loading control.

**Figure 4 metabolites-09-00188-f004:**
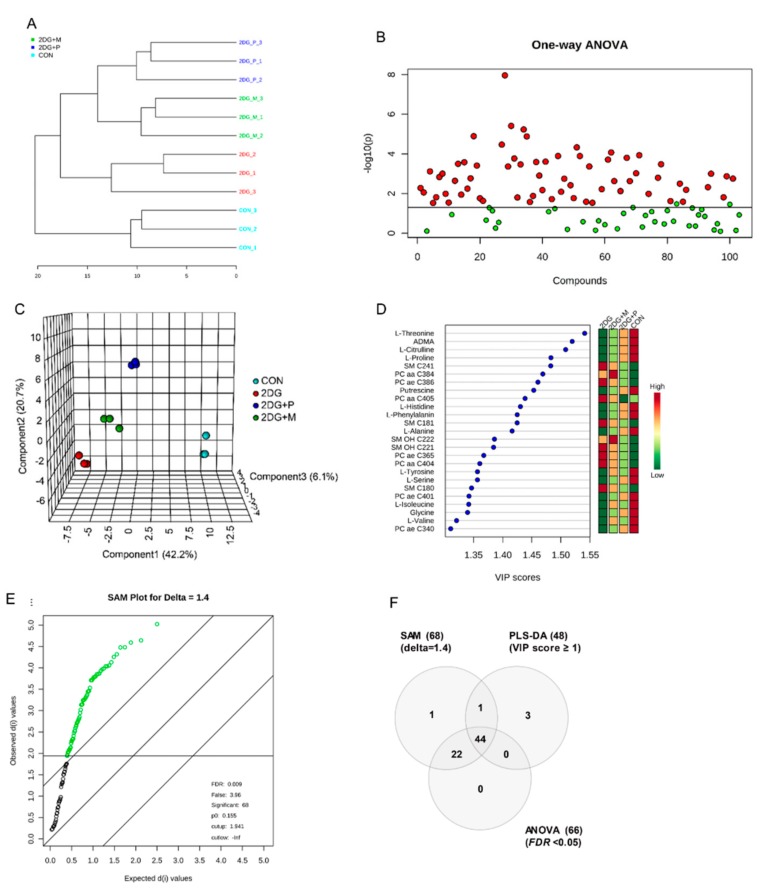
Statistical analysis for the identification of the altered metabolites in SCC15 cells. (**A**) Dendrogram of unsupervised hierarchical clustering for potential metabolites (distance measurement using Pearson’s correlation, and clustering algorithm using Ward. D). (**B**) ANOVA plot showing the important metabolites by one-way ANOVA and posthoc analysis (q value < 0.05, Tukey’s HSD) (**C**) 3D-Partial least squares-discriminant analysis (PLS-DA) scores scatter plot showing spatial division among all the four groups, R2X = 0.91, R2Y = 0.97, Q2X = 0.82, Q2Y = 0.92. (**D**) Key metabolites based on variable importance in projection (VIP) score plot in PLS-DA analysis. (**E**) Significant metabolites identified by SAM plot (Delta score = 1.4, green circles represent the significant metabolites). (**F**) Venn diagram showing the numbers of significantly altered metabolites identified using PLS-DA, ANOVA, and SAM methods.

**Figure 5 metabolites-09-00188-f005:**
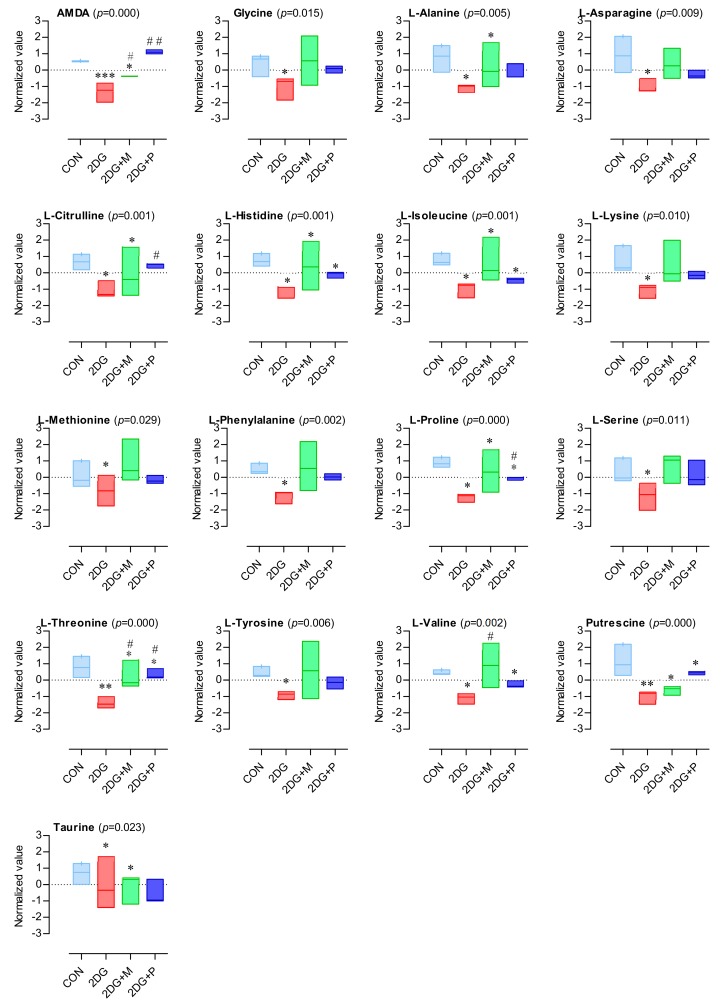
Box plot of altered amino acid and biogenic amine levels. Normalized values of all the experimental groups were determined by the MetaboAnalyst, a web-based platform (normalization by median, auto-scaling). Significant differences among the groups were assessed using one-way ANOVA (*p*-value < 0.05, Tukey’s HSD). The box plots were generated using GraphPad Prism; One-way ANOVA with Tukey’s multiple comparison test; ***: *p* < 0.001, **: *p* < 0.01, *: *p* < 0.05 (vs CON); ##: *p* < 0.01, #: *p* < 0.05 (vs 2DG).

**Figure 6 metabolites-09-00188-f006:**
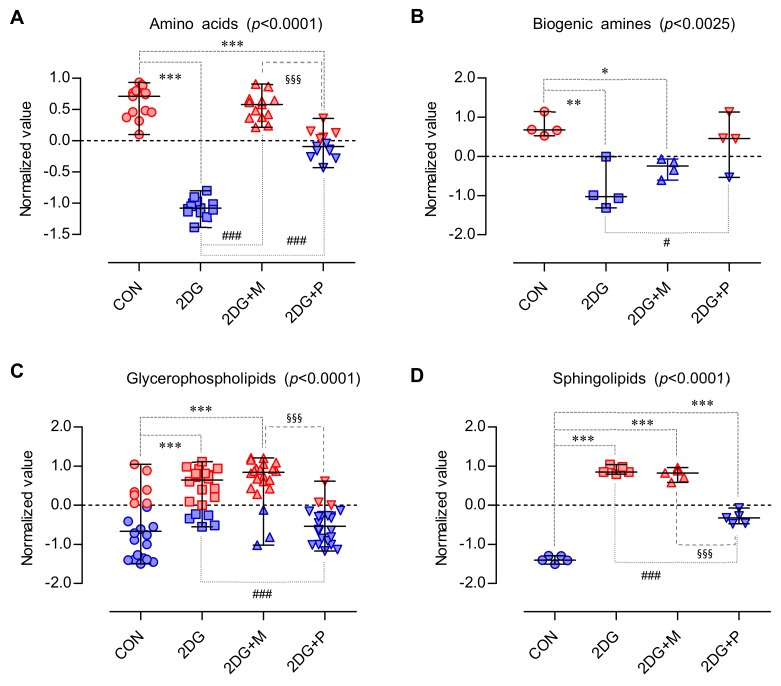
Scattered dot plot of the normalized values of the metabolites. The normalized values > 0 is marked in red and those < 0 in blue. (**A**) Amino acids (**B**) Biogenic amines (**C**) Glycerophospholipids (**D**) Sphingolipids; Scatter dot plot (median with range); One-way ANOVA with Tukey’s multiple comparison test; ***: *p* < 0.001, **: *p* < 0.01, *: *p* < 0.05 (vs CON); ###: *p* < 0.001, #: *p* < 0.05 (vs 2DG); §§§: *p* < 0.001 (vs 2DG + M).

**Figure 7 metabolites-09-00188-f007:**
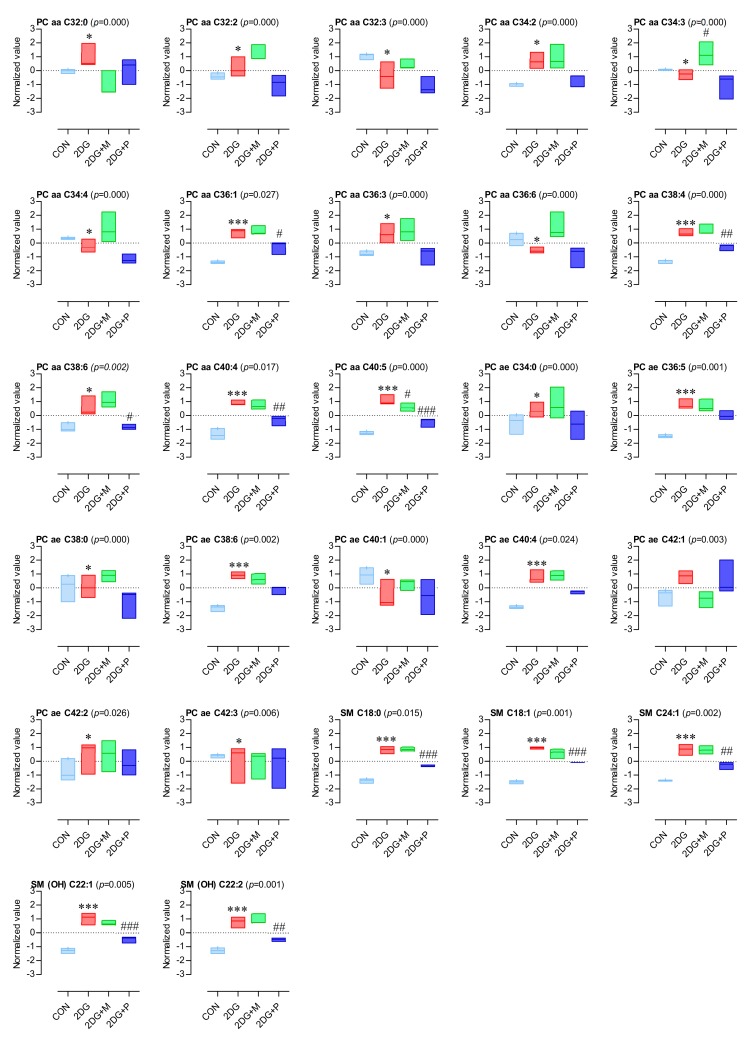
Box plot of altered lipid levels. The normalized values of all the experimental groups were determined by the MetaboAnalyst, a web-based platform (normalization by median, auto-scaling). Significant differences among the groups were assessed using one-way ANOVA (*p*-value < 0.05, Tukey’s HSD). The box plots were generated using GraphPad Prism; One-way ANOVA with Tukey’s multiple comparison test; ***: *p* < 0.001, *: *p* < 0.05 (vs CON); ###: *p* < 0.001, ##: *p* < 0.01, #: *p* < 0.05 (vs 2DG).

**Table 1 metabolites-09-00188-t001:** List of altered metabolites by PLS-DA, ANOVA and SAM analyses.

Class	Metabolite	KEGG ID	Normalized Value				*p*-Value	FDR	*d*-Value	VIP Score
			CON	2DG	2DG+M	2DG+P				
Amino Acids and Biogenic Amines	ADMA	C03626	0.53	−1.31	−0.35	1.13	0.000	0.000	4.482	1.520
Amino Acids and Biogenic Amines	Glycine	C00037	0.37	−1.00	0.58	0.05	0.015	0.029	2.340	1.340
Amino Acids and Biogenic Amines	l-Alanine	C00041	0.74	−1.08	0.21	0.13	0.005	0.013	2.816	1.416
Amino Acids and Biogenic Amines	l-Asparagine	C00152	0.93	−1.01	0.36	−0.28	0.009	0.019	2.595	1.303
Amino Acids and Biogenic Amines	l-Citrulline	C00327	0.67	−1.07	−0.06	0.46	0.001	0.003	3.534	1.508
Amino Acids and Biogenic Amines	l-Histidine	C00135	0.76	-1.09	0.42	−0.09	0.001	0.005	3.310	1.431
Amino Acids and Biogenic Amines	l-Isoleucine	C00407	0.77	−0.97	0.63	−0.43	0.001	0.004	3.448	1.342
Amino Acids and Biogenic Amines	l-Lysine	C00047	0.71	−1.05	0.48	−0.14	0.010	0.021	2.523	1.308
Amino Acids and Biogenic Amines	l-Methionine	C00073	0.10	−0.80	0.87	−0.16	0.029	0.046	2.041	1.229
Amino Acids and Biogenic Amines	l-Phenylalanine	C00079	0.48	−1.15	0.64	0.03	0.002	0.007	3.149	1.425
Amino Acids and Biogenic Amines	l-Proline	C00148	0.89	−1.22	0.37	−0.05	0.000	0.002	3.801	1.483
Amino Acids and Biogenic Amines	l-Serine	C00065	0.32	−1.13	0.67	0.15	0.011	0.023	2.478	1.357
Amino Acids and Biogenic Amines	l-Threonine	C00188	0.79	−1.39	0.24	0.35	0.000	0.002	3.850	1.541
Amino Acids and Biogenic Amines	l-Tyrosine	C00082	0.45	−0.90	0.61	−0.16	0.006	0.014	2.784	1.357
Amino Acids and Biogenic Amines	l-Valine	C00183	0.46	−1.10	0.90	−0.26	0.002	0.005	3.256	1.321
Amino Acids and Biogenic Amines	Putrescine	C00134	1.14	−0.99	−0.60	0.45	0.000	0.002	3.741	1.454
Amino Acids and Biogenic Amines	Taurine	C00245	0.69	0.00	−0.15	−0.53	0.023	0.041	2.139	1.067
Glycerophospholipids	PC aa C32:0	C00157	−0.04	0.98	−1.02	0.08	0.000	0.000	5.023	1.218
Glycerophospholipids	PC aa C32:2	C00157	−0.41	0.21	1.21	−1.01	0.000	0.000	4.643	1.208
Glycerophospholipids	PC aa C32:3	C00157	1.05	−0.34	0.42	−1.13	0.000	0.001	3.966	1.187
Glycerophospholipids	PC aa C34:2	C00157	−1.00	0.72	0.92	−0.64	0.000	0.002	3.779	1.125
Glycerophospholipids	PC aa C34:3	C00157	0.05	−0.25	1.21	−1.01	0.000	0.000	4.592	1.251
Glycerophospholipids	PC aa C34:4	C00157	0.34	-0.23	1.05	−1.17	0.000	0.000	4.478	1.196
Glycerophospholipids	PC aa C36:1	C00157	−1.39	0.78	0.88	−0.27	0.027	0.043	2.077	1.295
Glycerophospholipids	PC aa C36:3	C00157	−0.76	0.68	0.93	−0.85	0.000	0.002	3.858	1.013
Glycerophospholipids	PC aa C36:6	C00157	0.26	−0.52	1.16	−0.91	0.000	0.002	3.871	1.246
Glycerophospholipids	PC aa C38:4	C00157	−1.40	0.75	0.93	−0.28	0.000	0.001	4.037	1.469
Glycerophospholipids	PC aa C38:6	C00157	-0.88	0.61	1.09	−0.81	0.002	0.005	3.241	1.073
Glycerophospholipids	PC aa C40:4	C00157	−1.36	0.93	0.75	−0.32	0.017	0.031	2.290	1.361
Glycerophospholipids	PC aa C40:5	C00157	−1.27	1.11	0.61	−0.45	0.000	0.001	4.256	1.439
Glycerophospholipids	PC ae C34:0	-	−0.55	0.40	0.82	−0.67	0.000	0.001	3.932	1.311
Glycerophospholipids	PC ae C36:5	-	−1.50	0.81	0.68	0.01	0.001	0.004	3.463	1.368
Glycerophospholipids	PC ae C38:0	-	0.05	0.09	0.87	−1.01	0.000	0.001	4.057	1.287
Glycerophospholipids	PC ae C38:6	-	−1.45	0.94	0.65	−0.14	0.002	0.005	3.280	1.461
Glycerophospholipids	PC ae C40:1	-	0.88	−0.55	0.28	−0.62	0.000	0.002	3.788	1.342
Glycerophospholipids	PC ae C40:4	-	−1.39	0.78	0.90	−0.29	0.024	0.041	2.127	1.288
Glycerophospholipids	PC ae C42:1	-	−0.61	0.81	−0.81	0.62	0.003	0.009	3.020	1.068
Glycerophospholipids	PC ae C42:2	-	−0.71	0.43	0.44	−0.15	0.026	0.043	2.090	1.167
Glycerophospholipids	PC ae C42:3	-	0.39	0.00	−0.12	−0.27	0.006	0.015	2.726	1.230
Sphingolipids	SM C18:0	C00550	−1.41	0.85	0.88	−0.32	0.015	0.029	2.336	1.347
Sphingolipids	SM C18:1	C00550	−1.50	0.98	0.59	−0.07	0.001	0.005	3.340	1.425
Sphingolipids	SM C24:1	C00550	−1.40	0.85	0.83	−0.28	0.002	0.005	3.246	1.483
Sphingolipids	SM (OH) C22:1	C00550	−1.29	1.04	0.71	−0.46	0.005	0.012	2.854	1.384
Sphingolipids	SM (OH) C22:2	C00550	−1.29	0.80	0.96	−0.47	0.001	0.004	3.445	1.386

*p*-value: one-way ANOVA with Tukey’s multiple comparison test; FDR: false discovery rate; *d*-value: observed *d*(*i*) value of SAM, relative difference; VIP: variable importance in projection.

**Table 2 metabolites-09-00188-t002:** List of primer sequences used in PCR analysis.

Target Gene	Primers	Nucleotide Sequence
LDHA	F	GTGGGTCCTTGGGGAACATGGAG
	R	CAGGTTATCGGGTCCTACACATCGG
LDHB	F	CCGTCAGCAAGAAGGGGAGAGTCG
	R	GGTTAGTGCGCCACAAACCCATTAGG
HIF-1α	F	ACCACCTATGACCTGCTTGGTGCTG
	R	GTATAGGTCCGACACAGCTGACTCC
GAPDH	F	TCGACAGTCAGCCGCATCT
	R	CCGTTGACTCCGACCTTCA

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
