# Peer review of "2-Deoxy-d-Glucose-Induced Metabolic Alteration in Human Oral Squamous SCC15 Cells: Involvement of N-Glycosylation of Axl and Met"

_metabolites, 2019, doi:10.3390/metabo9090188_

Round 1

Reviewer 1 Report

Comments:

1) since this is a metabolomics journal and not a cancer journal please add a few sentences about Met and Axl in introduction.

2) Table 1:  Add information about p-value, FDR, d-value below table.

3) Figure 5 and 6 have captions about one-way ANOVA p-value but plots with significant p-values are not shown in the Figures. 

Reviewer 2 Report

Interesting paper, nicely presented, I did not find anything to criticize.

The authors tested a potent inhibitor of glycolysis, 2-deoxy-D-glucose (2DG) has been proposed for cancer treatment in human oral squamous SCC15 cells. They found that 2DG more effectively reduced glucose consumption and lactate level in SCC15 cells than in SCC4 cells, which are less dependent on glycolysis. Coincidentally, 2DG impaired N-linked glycosylation of the key oncogenic receptors Axl and Met in SCC15 cells, thereby reducing the cell viability and colony formation ability. This is relevant and interesting, because it explains the mechanism of how this drug works.

Reviewer 3 Report

In this manuscript, Lee et al. explore  how levels of individual proteins – in this case, receptor tyrosine kinases Axl and Met, which play an important role in cancer invasiveness and therapy resistance – can be modulated by perturbations to the cellular nutrient supply. Specifically, the authors find that 2-DG treatment reduces glycosylation and total levels of Axl and Met proteins in a manner that can be rescued by mannose but not pyruvate, which allows the authors to conclude that glycosylation impairment rather than the bioenergetic stress is the culprit. The authors also show that mannose but not pyruvate increases viability and colony formation of 2-DG treated cells. In the second half of the paper, Lee et al. report the data on the comprehensive metabolite profiling of 2-DG-treated cells, and uncover that 2-DG treatment leads to a decrease in virtually all amino acids in a manner that can be rescued by mannose and to a lesser extent, pyruvate. Our comments are outlined below.

The authors show convincingly that Axl and Met glycosylation and total levels are depleted by 2-DG treatment. However, the paper falls short of demonstrating the importance of this depletion for the effects on downstream signaling, cell viability and colony formation. Indeed, one would expect that 2-DG treatment would affect glycosylation of multiple transmembrane proteins, including other receptor tyrosine kinases. Given that the abstract states “2DG impaired N-linked glycosylation of the key oncogenic receptors Axl and Met in SCC15 cells, thereby reducing the cell viability and colony formation ability”, the authors need to take further steps to demonstrate the involvement of these particular kinases in the viability/colony formation phenotypes described. For example, testing whether the pharmacological inhibition of the kinase activity and/or RNAi/CRISPR-mediated depletion of Axl and Met would be sufficient to produce these phenotypes in SCC15 cells needs to be performed. Earlier papers linking glucose deprivation to expression levels of individual receptor tyrosine kinases, such as Wellen et al, Genes Dev, 2010 (PMID: 21106670) need to be referenced. The second half of the paper appears too descriptive and is disconnected from the first half. It is not clear whether the metabolic alterations the authors observe are linked to glycosylation impairment. Though the authors do not measure L-glutamine levels in this metabolite panel, one possibility that could explain the decrease in free amino acid levels across the board is a potential impairment in ASCT2 amino acid transporter, which imports glutamine - which, in turn, drives both the uptake of essential amino acids via LAT1 amino acid exchanger as well as serves as a substrate for non-essential amino acid synthesis. ASCT2 is known to be glycosylated and its glycosylation is essential for its membrane localization (see for example Console et al, Biochim Biophys Acta. 2015 (PMID: 25862406)). The authors could test whether ASCT2 and or LAT1 levels are impaired by 2-DG as a potential mechanistic explanation for a drop in amino acid levels observed.

In conclusion, the authors investigate an interesting problem and present clear and intriguing effects of 2-DG treatment. However, the conclusions of the paper need to be further backed by mechanistic data, discussion of the prior studies of the topic and a better cohesion between the first and the second half.

Round 2

Reviewer 3 Report

In the revised version of the manuscript, the authors have thoroughly addressed our comments. In particular, they have included new data on the sufficiency of Axl/Met inhibition for disruption of cell viability, using both small molecule inhibitors and siRNAs. The authors have also expanded the discussion of the results of the metabolite profiling presented in the second half of the paper to improve the conceptual cohesion between the first and second half of the manuscript. With these valuable additions, we now recommend this manuscript to be published in Metabolites journal.